# Suicide-Related Single Nucleotide Polymorphisms, *rs4918918* and *rs10903034*: Association with Dementia in Older Adults

**DOI:** 10.3390/genes13112174

**Published:** 2022-11-21

**Authors:** Olga Abramova, Kristina Soloveva, Yana Zorkina, Dmitry Gryadunov, Anna Ikonnikova, Elena Fedoseeva, Marina Emelyanova, Aleksandra Ochneva, Nika Andriushchenko, Konstantin Pavlov, Olga Pavlova, Valeriya Ushakova, Timur Syunyakov, Alisa Andryushchenko, Olga Karpenko, Victor Savilov, Marat Kurmishev, Denis Andreuyk, Olga Gurina, Vladimir Chekhonin, Georgy Kostyuk, Anna Morozova

**Affiliations:** 1Mental-Health Clinic No. 1 Named after N.A. Alekseev, Zagorodnoe Highway 2, 115191 Moscow, Russia; 2Department of Basic and Applied Neurobiology, V. Serbsky Federal Medical Research Centre of Psychiatry and Narcology, Kropotkinsky per. 23, 119034 Moscow, Russia; 3Center for Precision Genome Editing and Genetic Technologies for Biomedicine, Engelhardt Institute of Molecular Biology, Russian Academy of Sciences, 119991 Moscow, Russia; 4Department of Biology, Shenzhen MSU-BIT University, Ruyi Rd. 299, Shenzhen 518172, China; 5M.V. Lomonosov Moscow State University, 119991 Moscow, Russia; 6International Centre for Education and Research in Neuropsychiatry (ICERN), Samara State Medical University, 443016 Samara, Russia; 7Department of Medical Nanobiotechnology, Pirogov Russian National Research Medical University, 117997 Moscow, Russia; 8Federal State Budgetary Educational Institution of Higher Education “Moscow State University of Food Production”, Volokolamskoye Highway 11, 125080 Moscow, Russia

**Keywords:** dementia, cognitive impairment, suicide, single-nucleotide polymorphism, APOE, *rs429358*, *rs7412*, *rs4918918*, *rs10903034*

## Abstract

Dementia has enormous implications for patients and the health care system. Genetic markers are promising for detecting the risk of cognitive impairment. We hypothesized that genetic variants associated with suicide risk might significantly increase the risk of cognitive decline because suicide in older adults is often a consequence of cognitive impairment. We investigated several single-nucleotide polymorphisms that were initially associated with suicide risk in dementia older adults and identified the *APOE* gene alleles. The study was performed with subjects over the age of 65: 112 patients with dementia and 146 healthy volunteers. The MMSE score was used to assess cognitive functions. Study participants were genotyped using real-time PCR (*APOE*: *rs429358*, *rs7412;* genes associated with suicide: *rs9475195*, *rs7982251*, *rs2834789*, *rs358592*, *rs4918918*, *rs3781878*, *rs10903034*, *rs165774*, *rs16841143*, *rs11833579 rs10898553*, *rs7296262*, *rs3806263*, and *rs2462021*). Genotype analysis revealed the significance of *APOEε4*, *APOEε2*, and *rs4918918* (*SORBS1*) when comparing dementia and healthy control groups. The association of *APOEε4*, *APOEε2*, and *rs10903034* (*IFNLR1*) with the overall MMSE score was indicated. The study found an association with dementia of *rs4918918* (*SORBS1*) and *rs10903034* (*IFNLR1*) previously associated with suicide and confirmed the association of *APOEε4* and *APOEε2* with dementia.

## 1. Introduction

Neurocognitive disorders, especially severe dementia, have enormous implications for patients, their families, the health care system, and the economy. Approximately 50 million people suffer from dementia. This number is expected to triple by 2050 [1,2,3]. When dementia develops, it primarily affects memory and other cognitive functions, making daily life activities challenging and affecting orientation in time and space and speech [4]. The costs of disability caused by dementia account for a significant portion of the economic burden. In addition, family members of dementia patients suffer from increased emotional stress and depression, which affect their ability to work and contribute to the global cost of neurocognitive impairment [2]. Dementia can be caused by a wide range of diseases and lesions in the brain. Some common degenerative dementias in the elderly are Alzheimer’s disease (AD), dementia with Lewy bodies, vascular dementia, frontotemporal lobar degeneration, and Parkinson’s disease [5].

The most effective interventions that could delay or stop the progression of neurocognitive decline would be those initiated during the mild cognitive impairment phase or before the onset of any symptoms. Preventive therapy could significantly delay the onset of the disease, which in turn would improve the patient’s quality of life [6,7]. However, there are still no ways to diagnose the onset of cognitive decline. One such way might be to identify various biomarkers that could detect the onset of cognitive decline with a high degree of probability. Finding biomarkers would not only help to identify the risk of developing dementia but would also be crucial for monitoring the effects of treatment, which is also important for designing therapy. It could also help to objectify the effectiveness of new treatments in the long term [8,9]. However, there are currently no biomarkers recommended for clinical use.

Genetic markers are promising for detecting the risk of cognitive impairment because overall cognitive ability is substantially influenced by genetic factors since it depends on changes in chromosomes, genes, or proteins. Various mechanisms that have genetic components are thought to underlie pathological cognitive changes. The heritability of AD with late manifestation is 58-79%, while for AD with early manifestation, it is over 90%. Collaborative research by many scientific groups has accumulated a wealth of data on the genetics of AD [10,11] and has identified multiple loci associated with the disease [12,13]. The most studied, but not the most comprehensive, genetic risk factor for AD is the presence of the ε4 allele of the apolipoprotein E (*APOE*) gene in an individual. This allele is found in 20–25% of patients with AD, increasing the risk of developing the disease threefold in heterozygous carriers and 15-fold in homozygous carriers [13,14]. In addition, genome-wide association studies (GWAS) using samples from nearly 300,000 people have successfully identified more than 100 genome-wide significant loci associated with cognitive functions [15,16]. These functions include neurogenesis, regulation of nervous system development, neuronal differentiation, and regulation of cell development [16]. However, identifying single genes that could predict cognitive impairment with a high probability is challenging. A likely explanation is that dementia is a multifactorial disease in which multiple genes with small individual effects contribute to overall risk. Single-nucleotide polymorphisms (SNPs) alone generally have a small effect on disease risk and cannot be used as independent prognostic markers. Nevertheless, it is hypothesized that a combination of certain genes could be quite successful in identifying people at high risk of developing neurodegenerative diseases, but this requires the identification of risk genes.

In our work, we investigated the relationship between several SNPs that were initially associated with suicide risk and indicators of cognitive decline in dementia. We hypothesized that these genes might affect the overall risk of cognitive decline because suicide in older adults is often caused by mental illnesses, including depression, anxiety disorder, insomnia, AD, and vascular dementia. All these conditions are closely related to cognitive impairment, which plays a role in suicide attempts among older adults. The loss of some cognitive abilities in aging (dysfunctional cognitive control, executive function, and problem-solving) is a risk factor; such impairments impede coping with daily life problems and thus increase the risk of suicide. Overall, the extent to which cognitive aging explains the higher suicide rate in older adults remains unclear, but those diagnosed with dementia are known to be at a significantly higher risk of suicide than their healthy peers, having a 54% increased risk within the first year of diagnosis [17,18,19]. On a total sample of 594,674 patients, it was shown that patients younger than 65 and within 3 months of diagnosis had a 6.69 times higher risk of suicide than patients without dementia [20]. For this reason, it is necessary to consider the factor of increased risk of suicide when studying the features of cognitive impairment in aging. Presumably, the genetic factor may also have some significance, which is yet to be evaluated.

Suicidal thoughts, suicide attempts, and suicide are known to be largely inherited phenotypes [21]. There is evidence that the risk of suicide is increased fivefold in offspring whose parents have a history of suicide attempts [22]. Twin studies have estimated an inheritance rate of 17-55% for suicide and suicide attempts [23,24], while SNP-based estimates of suicide attempt inheritance were much more modest at 2–6% [25,26]. This heterogeneity of data indicates that the genetic epidemiology of suicide remains poorly understood [27]. In addition, there are currently few studies examining biomarkers of suicide risk in older adults, so this is a current challenge [27].

We suggested that genetic factors associated directly with suicide risk might also play a role in cognitive impairment in the elderly. We found it interesting to study the polymorphisms associated with suicide in a dementia cohort. In our study, we selected some SNPs that have been associated with suicidal behavior in earlier studies—*rs9475195*, *rs7982251*, *rs3781878*, *rs165774* [28], *rs2834789*, *rs10898553* [29], *rs358592* [30], *rs4918918*, *rs2462021* [31], *rs10903034* [32], *rs16841143* [26], *rs11833579* [33], *rs7296262* [34], and *rs3806263* [35]—and assessed the frequency of these genotypes in older adults diagnosed with dementia compared with their healthy peers. We selected polymorphisms that showed significant associations with suicide according to the GWAS as well as some other selected studies and that had a minor allele frequency in the European population of at least 10%. We evaluated the association between these polymorphisms and the index of cognitive impairment as well. In addition, we identified *APOE* polymorphisms in the study participants because their association with dementia risk is well known.

## 2. Materials and Methods

### 2.1. Participants

Patients with dementia were recruited in the gerontological department of Mental Health Clinic No. 1, named after N.A. Alekseev. Healthy volunteers were recruited in the period of 2020-2021 among volunteers who came for periodic medical examinations at Polyclinic No. 121 (Moscow). Informed consent was obtained from all participants. The study was conducted according to the guidelines of the Declaration of Helsinki. The procedures involving experiments on human subjects were performed in accordance with the ethical standards of Protocol No. 5, dated 20 September 2020, of the Ethics Committee of the Research Clinical Institute named after L.I. Sverzhevsky Moscow Healthcare Department. A total of 112 patients with dementia and 146 healthy volunteers took part in the study.

Inclusion criteria: all diagnoses were made at a daily consensus conference of neurologists, neuropsychologists, and psychiatrists. Diagnosis status was determined according to the International Classification of Diseases (ICD-10). All participants in the dementia and control groups underwent standardized neurological examination and neuropsychological testing. All subjects were assessed for subjective complaints of memory, normal general cognition, measurable impairment of one or more cognitive functions, and functional capacity, and daily performance was considered as well. The dementia group included subjects over 65 years of age with a diagnosis of dementia (F00—dementia with Alzheimer’s disease, F01—vascular dementia, F03—dementia unspecified). The control group included subjects over 65 years of age without cognitive impairment. Exclusion criteria for control and dementia groups: mental illness; positive family history—first-line relatives with mental illness; substance abuse, and severe comorbid somatic or neurological disorders.

### 2.2. Cognitive Status Assessment

The Mini-Mental State Examination (MMSE) was used to assess cognitive functions. The MMSE is a short 30-item questionnaire widely applied for the assessment and screening of cognitive impairments, including dementia. It is also used to evaluate the dynamics of cognitive functions against the background of ongoing therapy [36]. Subjects who scored 28–30 on the MMSE scale were considered healthy. Patients with dementia had a score of 24 or lower.

### 2.3. DNA Extraction and Genotyping

Genomic DNA was extracted from peripheral blood collected in EDTA-containing tubes using the DNA extraction kit (Syntol, LLC, Moscow, Russia) according to the manufacturer’s instructions.

The alleles of the *APOE* gene *(ε2*, *ε3*, *ε4)* were determined by real-time PCR based on genotyping for the *rs429358* and *rs7412* markers, as described previously [37], on the LightCycler^®^ 96 (Roche Diagnostics, Basel, Switzerland). Other markers were genotyped using the commercial TaqMan^®^ assay according to the manufacturer’s instructions on the QuantStudio™ 5 Real-Time PCR System (Thermo Fischer). The genes and polymorphisms chosen for analysis are described in Table 1.

### 2.4. Statistical Analysis

One-way analysis of variance was used to assess sociodemographic and clinical characteristics. Data were expressed as means ± SD. Compliance with the Hardy–Weinberg equilibrium was checked using the chi-square test. Statistical analysis was performed using the jamovi program (version 2.3.13.0). Associations between genetic markers and cognitive decline were analyzed using the SNPStats service (www.snpstats.net (accessed on 20 November 2022)) [42]. The dependent variables were the presence or absence of diagnosed dementia or MMSE scores. We used data from codominant, dominant, and recessive models. Differences were considered significant at *p* < 0.05. The *p*-values for the most significant models were corrected for multiple comparisons using the FDR procedure with the a priori threshold for statistical significance set to FDR < 0.05.

## 3. Results

Table 2 presents the characteristics of the samples studied, including healthy volunteers and patients with dementia. Analysis of variance revealed a significant difference between the groups in MMSE score (*p* < 0.001) and age (*p* < 0.001). Patients with dementia had lower MMSE scores and a higher age compared with healthy volunteers. The groups did not differ in education or the number of children.

In the study, 258 samples were genotyped for 16 genetic markers. The distribution of the genotypes is shown in Table 3.

Because of the difference in age between the groups, age was used as a covariate in the analysis using the SNPStats service. Significant associations were shown for several SNPs, but only *rs429358* was corrected for multiple comparisons (Table 4).

The *ε4* allele of the *APOE* gene was significantly associated with dementia in both heterozygous (OR = 2.23, 95% CI = 1.16–4.31, *p* = 0.001, FDR = 0.02) and homozygous (OR = 12.57, 95% CI = 1.41–112.41, *p* = 0.005, FDR = 0.08) carriage. In addition, significant results were shown for *rs7412* and *rs4918918*, which, however, did not pass the multiple comparison test. The *ε2* allele of the *APOE* gene was more common in the control group—significant associations were shown for heterozygotes in the codominant model (OR = 0.31, 95% CI = 0.12−0.81, *p* = 0.02, FDR = 0.16) and for homozygotes and heterozygotes compared with non-carriers (OR = 0.36, 95% CI = 0.14–0.91, *p* = 0.02, FDR = 0.19). For the *rs4918918* polymorphism in the *SORBS1* gene, significant associations were shown for homozygotes and heterozygotes compared with non-carriers in the dominant model (OR = 0.50, 95% CI = 0.26–0.95, *p* = 0.03, FDR = 0.18). The results of the genotype frequency analysis for all polymorphisms are shown in the Appendix A.

Analysis of the association between genotypes and the MMSE score showed several significant results, which nevertheless did not pass the multiple comparison test. The *APOE ε4* allele was associated with a lower MMSE score in homozygotes and heterozygotes compared with non-carriers in the dominant model (OR = −2.65, 95% CI = (−4.66)–(−0.63), *p* = 0.01, FDR = 0.09). By contrast, the *ε2* allele was associated with a higher MMSE score in heterozygotes compared with non-carriers in the codominant model (OR = 3.33, 95% CI = 0.60–6.07, *p* = 0.01, FDR = 0.07). For the *rs10903034* polymorphism in the *IFNLR1* gene, the homozygotes and heterozygotes were shown to be associated with a lower MMSE score compared with non-carriers in the dominant model (OR = −2.96, 95% CI = (−5.08)–(−0.84), *p* = 0.007, FDR = 0.11). The results of the analysis of the association between genotypes and MMSE scores for all polymorphisms are shown in the Appendix A.

## 4. Discussion

To date, there have been many genetic studies of dementia, including next-generation sequencing (NGS) studies, which contain much useful information on the association between dementia and various genetic variants [43,44]. However, there are currently no NGS studies on suicidal behavior, so it is of great interest to study this aspect, including in patients with dementia. Assessment of the relationship of SNPs, which are associated with the risk of suicide, with cognitive decline in older adults showed potential significance for *rs4918918* in the *SORBS1* gene and *rs10903034* in the *IFNLR1/IL10RB* gene. It should be noted that the results for these polymorphisms were not tested for multiple comparisons. However, we believe that attention should be paid to them because our study was performed on a small sample, while a larger-scale test could conceivably confirm the results.

The *SORBS1* gene is located on chromosome 10 and encodes a CBL-associated protein that is involved in insulin signaling (CAP/Ponsin protein, also known as Sorbin and SH3 domain-containing protein 1). SORBS1 is an important protein in the signaling pathway that mediates glucose uptake by the cell, which is stimulated by insulin. Genetic polymorphisms in this gene may regulate insulin resistance [45,46]. Insulin receptors are ubiquitous in the central nervous system. Insulin performs important functions in the brain, where it is involved in biological pathways relevant to neuronal survival, dendritic outgrowth, synaptic plasticity, and cognition, especially learning and memory [47]. Central insulin resistance is associated with various cognitive impairments and psychopathologies across the lifespan [47,48]. The expression product of the *SORBS1* gene is expressed in many tissues, most commonly adipose tissue, the heart, and other tissues, including the brain. Variations in the *SORBS1* gene are associated with glucose homeostasis, age of onset of diabetes, blood pressure, and high-density lipoprotein blood levels [45,49,50]. However, there are currently no studies demonstrating how the *rs4918918* polymorphism affects the expression of the *SORBS1* gene and through what mechanisms this polymorphism may affect brain function and behavior. Several studies support the potential role of the *SORBS1* gene in cognitive impairment. It was suggested that a genetic polymorphism in *SORBS1* is associated with cerebral infarction, which was shown in a Japanese population [48]. An epigenomic association study between cognitive ability and DNA methylation patterns in blood samples from monozygotic twins showed that the most associated finding of cognitive change was *SORBS1* [51]. In a meta-analysis using a large sample of 8737 patients, including 2805 patients with suicide attempts, *rs4918918* was shown to be associated with suicide attempts in patients with affective disorders [31]. Our study showed that the *rs4918918* polymorphism in the *SORBS1* gene differed in frequency between dementia and cognitively healthy individuals; the C allele presumably could be a risk factor for dementia. This further confirms earlier studies on the association between the *SORBS1* gene and cognitive dysfunction.

Another *rs10903034* polymorphism in our study was associated with scores on the MMSE scale (allele C was a risk factor). This polymorphism is located in the *IFNLR1* (Interferon Lambda Receptor 1) gene, which is localized in chromosome 1 and encodes the receptor for cytokine ligands, mediating their antiviral activity. The *IFNLR1* gene is expressed in many organs and tissues, including the cortex and cerebellum. Currently, there are few data on the functioning of this gene in brain tissues. However, it is known from animal experiments that there is an increase in the permeability of the blood–brain barrier in knockout mice by this gene, which indicates a potential role of this gene in brain function, although the exact mechanisms of the function are not understood [52]. Previous studies have not found a significant association between this gene and neurodegenerative diseases or psychiatric disorders, but a mutation at position *rs10903034* of this gene was associated with treatment-emergent suicidal ideation [32,53]. According to one theory, neuroinflammation can cause both neurodegeneration and mental dysfunction [54,55], and many immune dementia risk genes have already been identified [56]. The *rs10903034* polymorphism may be one of these genes, as our study showed a possible association between this genetic variant and cognitive dysfunction.

Our study showed a significant association between the *rs429358* polymorphism of the *APOE* gene and the diagnosis of dementia, which passed multiple comparison testing, with the C minor allele being a risk factor. The other *rs7412* polymorphism of the *APOE* gene showed significant associations (the C allele is a risk factor) without adjusting for multiple comparisons. In addition, both polymorphisms of the *APOE* gene showed an association with MMSE scores, which also did not pass the multiple comparison adjustment. The *APOE* gene encodes a multifunctional protein that plays a key role in lipoprotein metabolism both in the brain and in the periphery. In the periphery, ApoE is mainly synthesized by the liver and transports lipoproteins by binding to specific receptors. In the brain, ApoE is produced mainly by astrocytes and microglia and plays a key role in brain cholesterol metabolism and β-amyloid excretion [40,41,57]. The *APOE* gene is located on chromosome 19 in humans and is polymorphic, which means that it is characterized by a combination of polymorphisms 388T > C (*rs429358*) and 526C > T (*rs7412*). These polymorphisms form three major haplotypes (*ɛ2* (388T-526T), *ɛ3* (388T-526C), *ɛ4* (388C-526C)) and six genotypes (*ɛ2/ɛ2*, *ɛ2/ɛ3*, *ɛ2/ɛ4*, *ɛ3/ɛ3*, *ɛ3/ɛ4*, *ɛ4/ɛ4*) [40,58]. Lipids and plasma lipoproteins are strongly genetically influenced by the *APOE* polymorphism. Carriers of the *ε4* allele of the *APOE* gene (*APOE ε4*) have higher levels of total cholesterol than non-carriers. Therefore, *APOE ε4* is a strong genetic risk factor for cardiovascular disease [41]. In addition, *APOE ε4* is a major genetic risk factor for AD [39], which is further supported by a study that found differences between *APOE ε4* and *APOE ε3* in binding to β-amyloid [59]. In addition, the *APOE* polymorphism was shown to be associated with cognitive dysfunction in healthy individuals [38]. Our study confirmed the relationship of *APOE* gene variants with cognitive impairment in elderly people in our sample, as the risk alleles were more frequently found in the dementia group and were also associated with a lower score on the MMSE cognitive scale.

Our study has certain limitations. Suicide studies usually include subjects who have had suicide attempts or suicidal ideas. However, our study did not include this cohort of patients, which is a disadvantage because without this cohort, the associations identified have the risk of being false positives. For this reason, we need to be cautious about our results and further study the identified genetic variants in the cohort of patients with a suicidal history.

## 5. Conclusions

The results of our study confirmed the association of *rs429358* and *rs7412* polymorphisms of the *APOE* gene with cognitive impairment in older adults—it was shown that the risk alleles of these genetic variants are more common in older adults with dementia than in healthy individuals, and they are also associated with lower cognitive status scores. In addition, two genetic variants previously associated with suicide were identified for which a relationship with cognitive impairment in older adults was shown: *rs4918918* and *rs10903034*.

## Figures and Tables

**Table 1 genes-13-02174-t001:** Polymorphism descriptions.

SNP	Alleles	Gene	Assay *	Gene Product	Associations with Mental Disorders and Suicidal Behavior	References
*rs429358*	T > C	*APOE*	C_3084793_20	Apolipoprotein E (ApoE)	A risk factor for AD and cognitive dysfunction in healthy individuals	[38,39,40,41]
*rs7412*	C > T	*APOE*	C_904973_10			
*rs9475195*	T > C	*HCRTR2*	C_30334164_10	Orexin receptor type 2 (Ox2R)	Suicide attempts in bipolar disorder and schizophrenia	[28]
*rs7982251*	C > T	*FLT1*	C_29426110_10	Vascular endothelial growth factor receptor 1	Suicide attempts in bipolar disorder	[28]
*rs2834789*	T > C	*RUNX1*	C_16074160_10	α-Subunit of nuclear binding factor	Suicide attempts	[29]
*rs358592*	C > T	*KCNIP4*	C_8939728_10	Proteins that interact with potential-dependent potassium channels.	Suicidal ideation	[30]
*rs4918918*	T > C	*SORBS1*	C_3178855_10	CBL-associated protein that is involved in insulin signaling	Suicide attempts in affective disorders	[31]
*rs3781878*	G > A	*NCAM1*	C_2998862_10	Neural Cell Adhesion Molecule 1	Suicide attempts in bipolar disorder and depression	[28]
*rs10903034*	C > T	*IFNLR1*	C_2979476_10	Receptor for cytokine ligands IFNL2 and IFNL3	Suicidal thoughts in depressed patients	[32]
*rs165774*	G > A	*COMT*	C_2255325_10	Catechol-O-methyl transferase. Cytosolic enzyme that catalyzes the attachment of a methyl group to various catecholamines	Suicide attempts in bipolar disorder	[28]
*rs16841143*	G > A	*PTH2R*	C_2098753_10	Parathyroid hormone receptor type 2	Suicide attempts	[26]
*rs11833579*	G > A	*NINJ2*	C_1665834_10	Protein belonging to the ninjurin family, inducer of myelination	Ischemic stroke	[33]
*rs10898553*	T > C	*PRSS23*	C_1439661_10	Serine protease 23	Suicide attempts and suicidal thoughts among U.S. armed forces veterans	[29]
*rs7296262*	T > C	*TMEM132C*	C_1179719_10	Transmembrane protein 132C	Suicide attempts	[34]
*rs3806263*	G > A	*COQ8A (ADCK3)*	C_281882_10	Coenzyme Q8A is an atypical kinase involved in the biosynthesis of coenzyme Q	Suicide attempts in the Iranian Population	[35]
*rs2462021*	C > T	*JCAD*	C_202076_10	Protein functionally bound to cadherin 5	Suicide attempts in affective disorders	[31]

* TaqMan^®^ SNP Genotyping Assays.

**Table 2 genes-13-02174-t002:** Socio-demographic and clinical characteristics of healthy volunteers and dementia patients.

	Control	Dementia	*p*-Value
Number of participants	146 (56.6%)	112 (43.4%)	
Age, years	68.1 ± 6.3	76.7 ± 9.0	<0.001
Number of children	1.7 ± 0.6	1.4 ± 0.5	0.15
Education duration			
School education, years	9.8 ± 0.7	9.7 ± 0.7	0.74
Secondary education (including special education), years	11.2 ± 1.5	11.0 ± 1.6	0.57
Higher education, years	5.2 ± 1.6	4.7 ± 0.5	0.07
Assessment of cognitive functions			
Total MMSE score	28.9 ± 0.6	14.27 ± 6.7	<0.001

Data were expressed as Mean ± SD. A *p*-value < 0.05 was considered significant.

**Table 3 genes-13-02174-t003:** Distribution of alleles and test for Hardy–Weinberg equilibrium.

SNP	Control	Dementia	Test for Hardy–Weinberg Equilibrium (*p*-Value) *
APOE rs429358 (T > C)			
T/T	107 (75.9%)	65 (59.1%)	0.52
T/C	33 (23.4%)	37 (33.6%)	
C/C	1 (0.7%)	8 (7.3%)	
APOE rs7412 (C > T)			
C/C	121 (84%)	101 (91.8%)	1.00
C/T	23 (16%)	8 (7.3%)	
T/T	0 (0%)	1 (0.9%)	
HCRTR2 rs9475195 (T > C)			
T/T	40 (41.2%)	30 (30.9%)	0.30
C/T	42 (43.3%)	45 (46.4%)	
C/C	15 (15.5%)	22 (22.7%)	
FLT1 rs7982251 (C > T)			
T/T	94 (78.3%)	82 (76.6%)	0.75
C/T	24 (20%)	23 (21.5%)	
C/C	2 (1.7%)	2 (1.9%)	
RUNX1 rs2834789 (T > C)			
T/T	49 (46.2%)	41 (39.4%)	0.45
C/T	46 (43.4%)	45 (43.3%)	
C/C	11 (10.4%)	18 (17.3%)	
KCNIP4 rs358592 (C > T)			
T/T	47 (43.5%)	48 (50%)	0.51
C/T	52 (48.1%)	40 (41.7%)	
C/C	9 (8.3%)	8 (8.3%)	
SORBS1 rs4918918 (T > C)			
C/C	42 (40.4%)	57 (55.9%)	0.08
C/T	44 (42.3%)	35 (34.3%)	
T/T	18 (17.3%)	10 (9.8%)	
NCAM1 rs3781878 (G > A)			
G/G	54 (60%)	51 (51.5%)	1.00
A/G	32 (35.6%)	40 (40.4%)	
A/A	4 (4.4%)	8 (8.1%)	
IFNLR1 rs10903034 (C > T)			
T/T	46 (43%)	33 (31.7%)	0.47
C/T	44 (41.1%)	52 (50%)	
C/C	17 (15.9%)	19 (18.3%)	
COMT rs165774 (G > A)			
G/G	57 (54.8%)	48 (48.5%)	0.86
A/G	41 (39.4%)	42 (42.4%)	
A/A	6 (5.8%)	9 (9.1%)	
PTH2R rs16841143 (G > A)			
G/G	80 (79.2%)	86 (83.5%)	0.24
A/G	18 (17.8%)	16 (15.5%)	
A/A	3 (3%)	1 (1%)	
NINJ2 rs11833579 (G > A)			
G/G	52 (50%)	62 (60.2%)	0.58
A/G	45 (43.3%)	37 (35.9%)	
A/A	7 (6.7%)	4 (3.9%)	
PRSS23 rs10898553 (T > C)			
C/C	23 (30.3%)	22 (23.9%)	0.88
C/T	39 (51.3%)	47 (51.1%)	
T/T	14 (18.4%)	23 (25%)	
TMEM132C rs7296262 (T > C)			
T/T	33 (33.3%)	34 (32.1%)	0.89
C/T	49 (49.5%)	50 (47.2%)	
C/C	17 (17.2%)	22 (20.8%)	
COQ8A (ADCK3) rs3806263 (G > A)			
G/G	51 (52%)	54 (52.9%)	0.86
A/G	40 (40.8%)	39 (38.2%)	
A/A	7 (7.1%)	9 (8.8%)	
JCAD rs2462021 (C > T)			
T/T	63 (61.2%)	50 (51%)	0.71
C/T	35 (34%)	39 (39.8%)	
C/C	5 (4.8%)	9 (9.2%)	

* The test for Hardy–Weinberg equilibrium was performed for the total cohort.

**Table 4 genes-13-02174-t004:** Analysis of allele frequency association with study groups and association between genotypes and the MMSE score. Age was used as a covariate.

Genes	SNP	Association of Allele Frequency with Study Groups			Association of Genotypes with the MMSE Score		
		The Most Significant Model	*p*-Value	FDR	The Most Significant Model	*p*-Value	FDR
*APOE*	*rs429358*	Codominant	0.001	**0.02**	Dominant	0.01	0.09
*APOE*	*rs7412*	Codominant	0.02	0.16	Codominant	0.01	0.07
*HCRTR2*	*rs9475195*	Recessive	0.21	0.42	Recessive	0.08	0.34
*FLT1*	*rs7982251*	Dominant	0.29	0.39	Recessive	0.72	0.77
*RUNX1*	*rs2834789*	Recessive	0.21	0.37	Recessive	0.77	0.77
*KCNIP4*	*rs358592*	Dominant	0.25	0.40	Recessive	0.32	0.51
*SORBS1*	*rs4918918*	Dominant	**0.03**	0.18	Dominant	0.48	0.59
*NCAM1*	*rs3781878*	Dominant	0.08	0.30	Dominant	0.09	0.28
*IFNLR1*	*rs10903034*	Dominant	0.19	0.43	Dominant	0.007	0.11
*COMT*	*rs165774*	Recessive	0.11	0.35	Recessive	0.56	0.64
*PTH2R*	*rs16841143*	Dominant	0.53	0.65	Recessive	0.47	0.63
*NINJ2*	*rs11833579*	Dominant	0.16	0.43	Recessive	0.24	0.55
*PRSS23*	*rs10898553*	Recessive	0.28	0.41	Dominant	0.24	0.48
*TMEM132C*	*rs7296262*	Dominant	0.62	0.66	Dominant	0.36	0.52
*COQ8A (ADCK3)*	*rs3806263*	Dominant	0.75	0.75	Recessive	0.26	0.46
*JCAD*	*rs2462021*	Recessive	0.58	0.66	Recessive	0.12	0.32

FDR: false discovery rate; *p*-value < 0.05 was considered significant.

## Data Availability

Data available on request due to restrictions privacy.

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
