# Peer review of "Suicide-Related Single Nucleotide Polymorphisms, rs4918918 and rs10903034: Association with Dementia in Older Adults"

_genes, 2022, doi:10.3390/genes13112174_

Round 1
Reviewer 1 Report
This is a case-control study exploring the associations between genetic variants associated with suicide risk and APOE variants might significantly increase the risk of cognitive decline in 112 patients with dementia and 146 healthy volunteers. The study found an association with dementia of two SNPs previously linked to suicide: rs4918918 (SORBS1) and rs10903034 (IFNLR1).
I have several comments:
- Although the approach is based on a candidate SNP selection (and not a hypothesis-free approach) and has inherent limitations, this is actually how large-scale genomic research should find an application, i.e. associated variants in GWAS are tested in clinical samples. However, this is valid when the outcome is the same, in this case it should have been suicidality. This leads me to the first main concern, the fact the identified associations might be spurious (false positives). The authors should elaborate on this in the Discussion (limitations)
- Along this line, have they performed an a priori power analysis?
- There is no mention of quality controls for genetic analyses (HWE?)
Author Response
Dear Reviewer 1,
Thank you for taking the time and effort necessary to review our manuscript. We are grateful for analysis of our manuscript and for the helpful recommendations. We believe that your suggestions and questions have made a significant contribution to improving our work, and we would like to express our gratitude for this important contribution.
We took into consideration all of your recommendations and would like to present our corrections in more understandable way. Additions and corrections are highlighted in yellow; we hope this format will be convenient.
Answers to Reviewer's Points:
- Although the approach is based on a candidate SNP selection (and not a hypothesis-free approach) and has inherent limitations, this is actually how large-scale genomic research should find an application, i.e. associated variants in GWAS are tested in clinical samples. However, this is valid when the outcome is the same, in this case it should have been suicidality. This leads me to the first main concern, the fact the identified associations might be spurious (false positives). The authors should elaborate on this in the Discussion (limitations)
Response: We are grateful for your attention to this important issue. We fully agree with this statement, so we have added appropriate limitations to the text of the manuscript.
- Along this line, have they performed an a priori power analysis?
Response: We did not perform this analysis in our study.
- There is no mention of quality controls for genetic analyses (HWE?)
Response: The HWE analysis data are presented in Table 3.
Reviewer 2 Report
Authors found that rs4918918 (SORBS1) and rs10903034 (IFNLR1), suicide-related genes, are associated with cognitive impairment and they also confirmed the association of the APOEε4 and APOEε2 with dementia. This study might provide an important insight into the diagnoses and the understanding of the mechanism of dementia. The manuscript is well-written, but I have a few concerns to accept it for publication as follows.
1. The authors explain about SORBS1 and IFNLR1genes and the possible association with dementia and neurodegenerative diseases, but it is unclear how the SNPs of rs4918918 (SORBS1) and rs10903034 (IFNLR1) could affect on the molecular function or expression levels of SORBS1 and IFNLR1. Please discuss more.
2. Was there any case that the same patient had the SNPs of multiple genes (SORBS1, IFNLR1, APOEε4 and APOEε2)? If there was, please include the information and whether there is stronger association with MMSE score or not.
Author Response
Dear Reviewer 2,
We are grateful for the positive comments on our study and for taking the time and effort necessary to review our manuscript. We sincerely appreciate all valuable comments and suggestions, which helped us to improve the quality of our manuscript.
We took into consideration all of your recommendations and would like to present our corrections in more understandable way. We highlighted all corrections in the text of the manuscript in yellow. We hope that such form of answer will be appropriate.
Answers to Reviewer's Points:
1.The authors explain about SORBS1 and IFNLR1genes and the possible association with dementia and neurodegenerative diseases, but it is unclear how the SNPs of rs4918918 (SORBS1) and rs10903034 (IFNLR1) could affect on the molecular function or expression levels of SORBS1 and IFNLR1. Please discuss more.
Response: We are grateful for the interesting question, which gave us the opportunity to examine the information about the resulting genes in more detail. Of course, it is interesting to consider the specific mechanisms, but they are not known at this time. Nevertheless, the association of genes with disease needs to be studied, since the presented results of the association of genes with disease may draw the attention of the scientific community to the importance of these genetic factors, which may give impetus to additional research. We have discussed the presented genes in more detail, supplementing the discussions with interesting information.
2.Was there any case that the same patient had the SNPs of multiple genes (SORBS1, IFNLR1, APOEε4 and APOEε2)? If there was, please include the information and whether there is stronger association with MMSE score or not.
Response: This is quite ambiguous question as, no doubts, every subject in the study have these genes. Regarding association of polymorphisms of these genes with group factor, we conducted additional log-linear analysis and found no significant interactions between these specific genes. Furthermore, we tested ANOVA analysis of MMSE total score against interaction of any two genes you have mentioned and found no significant interaction, too. As we did not intended to find epistatic interactions, we think that these results may not to be reflected in the manuscript.
Round 2
Reviewer 1 Report
No further comments